# Variational Bayesian Inference for Nonlinear Hawkes Process with Gaussian Process Self-Effects

**DOI:** 10.3390/e24030356

**Published:** 2022-02-28

**Authors:** Noa Malem-Shinitski, César Ojeda, Manfred Opper

**Affiliations:** 1Institute of Mathematics, University of Potsdam, 14476 Potsdam, Germany; 2Artificial Intelligence Group, Technische Universität Berlin, 10623 Berlin, Germany; ojedamarin@tu-berlin.de (C.O.); manfred.opper@tu-berlin.de (M.O.); 3Centre for Systems Modelling and Quantitative Biomedicine, University of Birmingham, Birmingham B15 2TT, UK

**Keywords:** Bayesian inference, point process, Gaussian process

## Abstract

Traditionally, Hawkes processes are used to model time-continuous point processes with history dependence. Here, we propose an extended model where the self-effects are of both excitatory and inhibitory types and follow a Gaussian Process. Whereas previous work either relies on a less flexible parameterization of the model, or requires a large amount of data, our formulation allows for both a flexible model and learning when data are scarce. We continue the line of work of Bayesian inference for Hawkes processes, and derive an inference algorithm by performing inference on an aggregated sum of Gaussian Processes. Approximate Bayesian inference is achieved via data augmentation, and we describe a mean-field variational inference approach to learn the model parameters. To demonstrate the flexibility of the model we apply our methodology on data from different domains and compare it to previously reported results.

## 1. Introduction

Sequences of self-exciting, or inhibiting, temporal events are frequent footmarks of natural phenomena: earthquakes are known to be temporally clustered as aftershocks are commonly triggered following the occurrence of a main event [1]; in social networks, the propagation of news can be modeled in terms of information cascades over the edges of a graph [2]; and in neuronal activity, the occurrence of one spike may increase or decrease the probability of the occurrence of the next spike over some time period [3].

Traditionally, sequences of events in continuous time are modeled by point processes, of which Cox processes [4], or doubly stochastic processes, use a stochastic process for the intensity function, which depends only on time and is not effected by the occurrences of the events. The Hawkes process [5] extends the Cox process to capture phenomena in which the past events affect future arrivals, by introducing a memory dependence via a memory kernel, which is also referred to as the causal influence function. When incorporating the dependence of the process on its own history, due to the superposition theorem of the point process, new events will depend on either an exogenous rate, which is independent of the history, or an endogenous rate from past arrivals.

Originally, the dependence on history in the Hawkes process is assumed to be self-excitatory, and the memory kernel is parameterized by an exponential or power law decay, which results in a model with low flexibility. Furthermore, assuming only an excitatory relation between the events does not hold for other phenomena we wish to model. For example, inhibitory effects between neurons [6], and even self-inhibition [7], are crucial for regulating the neuronal activity. Thus, the memory kernel should also include inhibitory relations between the events and by doing so the intensity may become negative. To ensure that the intensity function is non-negative, a nonlinear link function is applied on the memory kernel, and the resulting model is often referred to as a nonlinear Hawkes process [8,9,10]. Theoretical results for nonlinear Hawkes processes have been developed for many years and they include stability estimates [8] as well as convergence rates for Bayesian estimators [11].

In this work, we present a Nonlinear Hawkes Process with Gaussian Process Self-Effects (NH–GPS), which extends the class of nonlinear Hawkes processes. As the causal influence between events can be either excitatory or inhibitory, we use the term *self-effects* to refer to the influence of past events on future events.

We choose a semi-parametric approach, which avoids the limiting parameterization of the memory kernel and the background rate. We assume a Gaussian process (GP) prior to the exogenous events’ intensity and on the memory kernel, which allows also for an inhibitory effect between the events. To ensure that the intensity function is non-negative we use the sigmoid link function. This modeling approach is not only descriptive but also allows us to obtain a fast inference procedure. The history of self-effects defines an aggregated Gaussian process, and we perform the inference directly on this aggregation rather than obtaining a posterior over each self-effect.

While highly flexible approaches to modeling the intensity function of nonlinear Hawkes processes have been presented before, they mainly rely on deep neural network solutions [12,13]. These approaches are hindered by the necessity of large datasets. In contrast, our methodology retains modeling flexibility due to the nonparametric nature of Gaussian processes, while being able to perform well when data are scarce.

### Outline

In Section 2, we describe the NH–GPS model and emphasize how its structure allows for efficient Bayesian inference. In Section 3, we describe the augmentation scheme and derive the mean-field variational inference algorithm. In Section 4, we discuss related work and describe how it relates to ours. In Section 5, we present the results of our experiments both on synthetic data and different real-world examples, and compare it to existing work when possible. In Section 6, we conclude by discussing our work and future research directions.

## 2. Proposed Model

### 2.1. Classical Hawkes Process

Let $T_{t}=\left[0,t\right]\in R$. We define the counting measure $N(T_{t})$ as the number of arrivals in the interval $T_{t}$. Furthermore, we define the history $H_{t}$ or the realizations of a given process, as the set of arrivals in the interval $T_{t}$, namely $H_{t}=\left\{T_{1},\ldots ,T_{N(T_{t})}:T_{i}\in T_{t}\wedge T_{i-1}<T_{i}\right\}$, and $T_{i}$ corresponds to the time of arrival *i*. For a temporal point process, the counting measure N(·) has an associated intensity defined as
$$\begin{array}{c} \Lambda \left(t\right)=\lim_{\Delta t\to 0}\frac{E[N\left(T_{t+\Delta t}\right)-N\left(T_{t}\right)|H_{t}]}{\Delta t}. \end{array}$$

The intensity function may depend on the history of the process. An example of such a process is the Hawkes process, or self-exciting point process [14], which defines self-excitations [15] around *exogenous events*. Following Hawkes and Oakes [5], the intensity of the Hawkes process is defined by
(1)$$\begin{array}{c} \Lambda \left(t|H_{t}\right)=s\left(t\right)+\sum_{t_{n}<t}g\left(t-t_{n}\right), \end{array}$$
where s(t) is the base intensity of exogenous arrivals and $g\left(t-t_{n}\right)$ is the memory kernel, or causal influence function, defining the change in the excitation or inhibition value following each arrival. In the classical Hawkes process, causal influence of only excitatory nature is allowed and the memory kernel is usually of the form $g\left(t-t_{n}\right)=\beta e^{-\alpha (t-t_{n})}$ for an exponentially decaying memory.

### 2.2. Nonlinear Hawkes Process with Gaussian Process Self-Effects

In the classical Hawkes process, the memory kernel *g* in Equation (Equation 1) must be non-negative, to prevent the intensity function from being negative. As a result the history of the model has only excitatory effect on future events. We are interested in a model that includes inhibition between events, and we release the constraint over *g* so it can be negative, and define the following nonlinear intensity function
(2)$$\Lambda \left(t\right)\;=\;\lambda \sigma \left(\phi (t)\right)$$
(3)$$\sigma \left(\phi (t)\right)\;=\;\frac{1}{1+\exp\left(-\phi \left(t\right)\right)}$$
(4)$$\phi \left(t\right)=s\left(t\right)\;+\;\sum_{t_{n}<t}g\left(t-t_{n}\right)\exp\left(-\alpha \left(t-t_{n}\right)\right).$$

Here, we choose the sigmoid function to ensure that the intensity function $\Lambda \left(\cdot \right)$ is non-negative. λ is the intensity bound and we refer to $\phi \left(\cdot \right)$ as the linear intensity function.

We explicitly add the exponential decay to enforce the forgetting constraint, which is essential for most realistic processes. Although we choose here a specific parameterization of the memory decay, one can choose other forms of memory decay with minimal adaptation to the learning procedure of the model parameters.

Besides α and λ, the functions s(·) and g(·) are the unknown parameters of the model to be inferred from the data. In this paper, we will use a nonparametric Bayesian inference approach based on the definition of a prior probability measure over these functions. In contrast to alternative Bayesian models for Hawkes processes where the positive rate function is directly modeled by a Dirichlet process, in our case we have to deal with random functions which are not constrained to be positive. This suggests that we should use a simple, but still highly flexible approach by modeling the two functions independently as realizations of Gaussian random processes. We write symbolically: (5)$$s\;∼\;GP\left(0,K^{s}\right)$$(6)$$g\;∼\;GP\left(0,K^{g}\right)$$(7)$$K^{s/g}\left(t_{1},t_{2}\right)\;=\;a_{s/g}\cdot \exp\left(-\frac{∥t_{1}-t_{2}{∥}^{2}}{\sigma_{s/g}^{2}}\right).$$

The corresponding Gaussian prior measures are uniquely defined by the mean functions (which we set to be equal to zero) and the second moments given by the covariance kernel functions $K^{g/s}$. The latter are defined by the prior expectations
(8)$$K^{s}\left(t_{1},t_{2}\right)\;≐\;E[s\left(t_{1}\right)s\left(t_{2}\right)]$$
(9)$$K^{g}\left(t_{1},t_{2}\right)\;≐\;E[g\left(t_{1}\right)g\left(t_{2}\right)]$$

By a proper choice of kernels, we can encode further prior beliefs on typical realizations of s(·) and g(·). Throughout the paper, we will work with the so-called RBF kernels from Equation (7). This kernel corresponds to the prior assumptions that the Gaussian processes are stationary (the kernels depend on time differences only) and that the functions s(·) and g(·) are infinitely often differentiable. The kernels depend on two hyperparameters *a* and σ, which reflect the typical amplitude and length scale of the functions. The reasonable values of these hyperparameters will also be inferred from the data.

Finally, we assume a prior distribution also on the upper intensity bound
$$\begin{array}{c} \lambda ∼Gamma\left(\alpha_{0},\beta_{0}\right). \end{array}$$
and we identify the hyperparameters of the model as $\left\{\sigma_{g},a_{g},\alpha ,\sigma_{s},a_{s}\right\}$.

In this work, we propose Bayesian inference for fitting the model to data. Due to the nonlinearity over $\phi \left(\cdot \right)$, we are no longer able to easily utilize the branching structure of the Hawkes process, which allowed for the estimation of $s\left(\cdot \right)$ and $g\left(\cdot \right)$ [16,17]. Thus, a natural solution is to perform the inference directly on $\phi \left(\cdot \right)$.

Next, we identify the prior over the entire linear intensity $p\left(\phi \right)$. From Equation (4) we see that the linear intensity function ϕ is nothing but the sum of GPs, and as such it is also a GP:(10)$$\begin{array}{cc} \; & \phi ∼GP\left(0,\tilde{K}\right) \end{array}$$(11)$$\begin{array}{cc} \; & {\tilde{K}}_{lk}=K_{lk}^{s}+\sum_{t_{i}<t_{l}}\sum_{t_{j}<t_{k}}K_{t_{l}-t_{i},t_{k}-t_{j}}^{g}\exp\left(-\alpha \left(t_{l}-t_{i}+t_{k}-t_{j}\right)\right). \end{array}$$

#### Multivariate Model

We propose an extension of the model to multiple dimensions. This is useful in applications where different types of events are observed, or the events originate from different processes that affect each other. We define an *R*-dimensional point process with intensity in dimension *r*
$$\begin{array}{cc} \; & \Lambda^{r}\left(t\right)=\lambda^{r}\sigma \left(\phi^{r}\left(t\right)\right)\\ \; & \phi^{r}\left(t\right)=s^{r}\left(t\right)+\sum_{m=1}^{R}\sum_{t_{n}^{m}<t}g_{r,m}\left(t-t_{n}^{m}\right)\exp\left(-\alpha_{r,m}\left(t-t_{n}^{m}\right)\right) \end{array}$$
where $t_{n}^{m}$ is the time of event number *n* of type *m*. We assume that every dimension has its own intensity bound $\lambda^{k}$ and background rate $s^{r}\left(\cdot \right)$. The different dimensions interact with each other via the self-effects term. $g_{r,m}\left(\cdot \right)$ defined the effects of the events of type *m* on the events of type *r*. As in the univariate case, this effect may change over time.

Given the observations, the different dimensions are independent of each other and we can learn their parameters separately. Thus, in the following section we present the inference for the univariate model, and the extension to the multivariate case is straight forward.

## 3. Inference

Conditioned on the intensity function $\Lambda \left(\cdot \right)$, the likelihood of observations $\left\{t_{1},\ldots t_{N}\right\}$ from a Hawkes process is [18]
$$\begin{array}{c} \ell \left(\left\{t_{1},\ldots t_{n}\right\}|\Lambda \left(\cdot \right)\right)=\exp\left\{-\int_{0}^{T}\Lambda \left(t^{\prime }\right)dt^{\prime }\right\}\prod_{n=1}^{N}\Lambda \left(t_{n}\right). \end{array}$$

In order to obtain posterior distributions for the latent variables either in the form of a Gibbs sampler or through approximate posteriors in variational inference, we require certain computations that are not tractable or efficient under the current form of the likelihood. Similarly to previous work on Cox and Hawkes processes [17,19,20], we follow an augmentation procedure. We do so via the introduction of auxiliary variables, which expand the model to a different likelihood form. Under the marginalization of the aforementioned auxiliary variables, the new likelihood will conserve the form of the original model likelihood. The new form of the likelihood is constructed such that the computations required for the inference procedure are either tractable, computationally fast, or both.

### 3.1. Model Augmentation

The first step we take in treating the likelihood function is using the Pólya–Gamma (PG) augmentation scheme. Following Theorem 1 in Polson et al. [21], we can rewrite the nonlinear intensity function as
(12)$$\begin{array}{cc} \; & \sigma \left(\phi \left(t\right)\right)=\int_{0}^{\infty }e^{f\left(w,\phi \left(t\right)\right)}PG\left(w;1,0\right)dw \end{array}$$
(13)$$\begin{array}{cc} \; & f\left(w,\phi \left(t\right)\right)=-\frac{\phi {\left(t\right)}^{2}w}{2}+\frac{\phi \left(t\right)}{2}-\ln2. \end{array}$$

As we augment each observation with a variable $w_{n}$ from a PG distribution, the joint likelihood of the observed events $\left\{t_{n}\right\}$ and PG variables $\left\{w_{n}\right\}$ is
(14)$$\begin{array}{cc} \; & p\left({\left\{t_{n}\right\}}_{n=1}^{N},{\left\{w_{n}\right\}}_{n=1}^{N}|\phi ,\lambda \right)=\\ \; & \exp\left(-\int_{0}^{T}\lambda \sigma \left(\phi \left(t\right)\right)dt\right)\cdot \prod_{n=1}^{N}\lambda e^{f\left(w_{n},t_{n}\right)}PG\left(w_{n};1,0\right) \end{array}$$
with
(15)$$\begin{array}{cc} \; & \exp\left\{-\int_{0}^{T}\lambda \sigma \left(\phi \left(t\right)\right)dt\right\}=\\ \; & \exp\left(-\int_{0}^{T}\int_{0}^{\infty }\lambda PG\left(w;1,0\right)\left(1-e^{f\left(w,-\phi \left(t\right)\right)}\right)dwdt\right). \end{array}$$
where we used σ(t)=1−σ(−t).

Next, we utilize the Campbell’s theorem [14], which states that for a Poisson process Π with intensity φ
$$\begin{array}{cc} E_{\varphi } & \left(\prod_{x\in \Pi }\exp\left(h\left(x\right)\right)\right)=\\ & \exp\left(-\int \left(1-\exp\left(h\left(x\right)\right)\right)\varphi \left(x\right)dx\right). \end{array}$$

Looking at Equation (Equation 15), we identify $x=\left(t,w\right)$ and $\varphi \left(t,w\right)=\lambda PG\left(w|1,0\right)$ is the intensity of a marked Poisson process in T with marks $w∼PG\left(0,1\right)$. Furthermore, we determine $h\left(x\right)=f\left(w,-\phi \left(t\right)\right)$. We can now rewrite the exponential in Equation (Equation 14) as
(16)$$\begin{array}{c} \exp\left\{-\int_{0}^{T}\lambda \sigma \left(\phi \left(t\right)\right)dt\right\}=E_{\varphi }\left(\prod_{m=1}^{M}e^{f\left({\hat{w}}_{m},{\hat{t}}_{m}\right)}\right) \end{array}$$
for realizations ${\left\{{\hat{t}}_{m},{\hat{w}}_{m}\right\}}_{m=1}^{M}$.

We substitute Equation (Equation 16) into Equation (Equation 14), which results in the full augmented likelihood. Given the prior distributions over ϕ and λ, we can now write the model’s posterior distribution as
(17)$$\begin{array}{cc} \; & p\left(\left\{{\hat{t}}_{m},{\hat{w}}_{m}\right\},\left\{w_{n}\right\},\phi ,\lambda |\left\{t_{n}\right\}\right)\propto \exp\left(-\lambda T\right)\times \\ \; & \prod_{m=1}^{M}\lambda e^{f\left({\hat{w}}_{m},-\phi \left({\hat{t}}_{m}\right)\right)}PG\left(w_{m};1,0\right)\times \\ \; & \prod_{n=1}^{N}\lambda e^{f\left(w_{n},\phi \left(t_{n}\right)\right)}PG\left(w_{n};1,0\right)\times p\left(\phi \right)p\left(\lambda \right). \end{array}$$

To summarize, we augment the model with two sets of variables—the PG variables $\left\{w_{n}\right\}$, which augment the actual realizations, and the tuples $\left\{{\hat{t}}_{m},{\hat{w}}_{m}\right\}$, which are the realizations and marks of the auxiliary marked Poisson process.

As mentioned above, we intend to learn directly the linear intensity function $\phi \left(\cdot \right)$. This allows us to utilize the mean-field variational inference previously introduced in Donner and Opper [19] and Donner and Opper [22]. Next, we go through the steps of the algorithm, and we refer the reader to the two papers mentioned above for further details. As a baseline we compare the performance of the variational inference algorithm to a Gibbs sampler. The details of the Gibbs sampler can be found in Appendix A and Algorithm 2.

### 3.2. Variational Inference

In the variational inference [23,24] we define a tractable distribution family and adapt it to approximate the posterior by maximizing the lower bound L(Q) defined below. This procedure minimizes the Kullback–Leibler divergence between the unknown posterior and the proposed approximating distribution. The posterior density is approximated by
$$\begin{array}{cc} \; & p\left(\left\{{\hat{t}}_{m},{\hat{w}}_{m}\right\},\left\{w_{n}\right\},\phi ,\lambda |\left\{t_{n}\right\}\right)\\ \; & \approx q_{1}\left(\phi ,\lambda \right)q_{2}\left({\left\{w_{n}\right\}}_{n=1}^{N},{\left\{{\hat{t}}_{m},{\hat{w}}_{m}\right\}}_{m=1}^{M}\right). \end{array}$$

This leads to the following lower bound on the evidence
$$\begin{array}{c} L\left(Q\right)=E_{Q}\left[\log\left\{\frac{p\left(\left\{{\hat{t}}_{m},{\hat{w}}_{m}\right\},\left\{w_{n}\right\},\phi ,\lambda |\left\{t_{n}\right\}\right)}{q_{1}\left(\phi ,\lambda \right)q_{2}\left({\left\{w_{n}\right\}}_{n=1}^{N},{\left\{{\hat{t}}_{m},{\hat{w}}_{m}\right\}}_{m=1}^{M}\right)}\right\}\right]. \end{array}$$

Here, *Q* refers to the probability measure of the variational posterior. We can maximize the bound by alternating the maximization over each of the factors [24]. The optimal solution for each factor is
(18)$$\begin{array}{cc} \; & \log q_{1}^{*}\left(\phi ,\lambda \right)=\\ \; & E_{q_{2}\left({\left\{w_{n}\right\}}_{n=1}^{N},{\left\{{\hat{t}}_{m},{\hat{w}}_{m}\right\}}_{m=1}^{M}\right)}\left[\log P\left(\left\{{\hat{t}}_{m},{\hat{w}}_{m}\right\},\left\{w_{n}\right\},\phi ,\lambda ,\left\{t_{n}\right\}\right)\right] \end{array}$$
(19)$$\begin{array}{cc} \; & \log q_{2}^{*}\left({\left\{w_{n}\right\}}_{n=1}^{N},{\left\{{\hat{t}}_{m},{\hat{w}}_{m}\right\}}_{m=1}^{M}\right)=\\ \; & E_{q_{1}\left(\phi ,\lambda \right)}\left[\log P\left(\left\{{\hat{t}}_{m},{\hat{w}}_{m}\right\},\left\{w_{n}\right\},\phi ,\lambda ,\left\{t_{n}\right\}\right)\right]. \end{array}$$

Thus, to obtain the optimal distribution of one of the factors, one must calculate the expectations of the logarithm of the joint distribution over the remaining factors in the approximation, resulting in an iterative algorithm.

In the following subsections, we explicitly express the functional form of the optimal distributions, and obtain the corresponding expectations required for updating the factors.

### 3.3. Optimal $Q_{1}$

We find that the optimal $q_{1}$ is factorized as
$$\begin{array}{c} q_{1}\left(\phi ,\lambda \right)=q_{1}\left(\lambda \right)q_{1}\left(\phi \right) \end{array}$$

The first factor is identified as a Gamma distribution
(20)$$\begin{array}{cc} \; & q_{1}\left(\lambda \right)=Gamma\left(\alpha ,\beta \right)\\ \; & \alpha =\alpha_{0}+N+\int_{TxW}\Lambda_{q_{2}}\left(t,w\right)dtdw\\ \; & \beta =\beta_{0}+T \end{array}$$
with known expectations.

The optimal distribution for the second factor is of the form
$$\begin{array}{cc} q_{1}^{*} & \propto e^{-U\left(\phi \right)+\log p\left(\phi \right)}\\ U(\phi ) & =\frac{1}{2}\int A\left(t\right)\phi^{2}\left(t\right)dt-\int b\left(t\right)\phi \left(t\right)dt\\ A(t) & =\sum_{n}{\langle \omega_{n}\rangle }_{q_{2}^{*}}\delta \left(t-t_{n}\right)+{\langle \omega \left(t\right)\rangle }_{q_{2}^{*}}\Lambda_{q_{2}^{*}}\left(t\right)\\ b(t) & =\sum_{n}\frac{1}{2}\delta \left(t-t_{n}\right)-\frac{1}{2}\Lambda_{q_{2}^{*}}\left(t\right). \end{array}$$

Generally, the integrals above cannot be evaluated analytically. Thus, we resort to another variational approximation, where we approximate the likelihood term, by a distribution that depends only on a finite set of inducing point {c}, $\tilde{q}\left(\phi_{c},\phi \right)=p\left(\phi |\phi_{c}\right)q\left(\phi_{c}\right)$, the ELBO is
$$\begin{array}{c} {\langle \log\frac{e^{-\log{\langle U\left(\phi \right)\rangle }_{p(\phi |\phi_{c})}}p\left(\phi_{c}\right)}{\tilde{q}\left(\phi_{c}\right)}\rangle }_{\tilde{q}} \end{array}$$
and we use the notation ${\langle p\rangle }_{q}=E_{q}\left(p\right)$. The optimal $\tilde{q}\left(\phi_{c}\right)$ is given by
$$\begin{array}{c} {\tilde{q}}^{*}\left(\phi_{c}\right)\propto e^{-\log{\langle U\left(\phi \right)\rangle }_{p(\phi |\phi_{c})}}p\left(\phi_{c}\right). \end{array}$$

From here, using the known results of conditional GPs and sparse variational GPs [25,26], we have
(21)$$\begin{array}{cc} {\tilde{q}}^{*}\left(\phi_{c}\right) & =N(\phi_{c}|\mu_{c},\Sigma_{c})\\ \Sigma_{c} & ={\left[\int \kappa {\left(t\right)}^{⊤}A\left(t\right)\kappa \left(t\right)dt+K_{c}^{-1}\right]}^{-1}\\ \mu_{c} & =\Sigma_{c}\left(\int b(t)\kappa (t)dt\right) \end{array}$$
with $K_{c}$ the covariance kernel between the inducing points, $\kappa \left(t\right)=k_{c}{\left(t\right)}^{⊤}K_{c}^{-1}$ and $k_{c}\left(t\right)$ is the kernel between the inducing points and another set of points (either the real data or the integration points), such that $k_{c}\left(t\right)=\left(\tilde{K}\left(t,t_{1}\right),\ldots ,\tilde{K}\left(t,t_{L}\right)\right)$ with $t_{1},\ldots ,t_{l},\ldots ,t_{L}$ are the inducing points. The mean and the variance of the sparse approximated GP are
(22)$$\begin{array}{cc} \langle \phi (t)\rangle = & \kappa \left(t\right)\mu_{c} \end{array}$$
(23)$$\begin{array}{cc} \sigma^{2}\left(\phi \left(t\right)\right)= & K\left(t,t\right)-\kappa {\left(t\right)}^{⊤}k_{c}\left(t\right)+\kappa {\left(t\right)}^{⊤}\Sigma_{c}\kappa \left(t\right) \end{array}$$

### 3.4. Optimal $Q_{2}$

Similarly to the previous section, we find that the optimal $q_{2}$ is factorized as
$$\begin{array}{c} q_{2}\left({\left\{w_{n}\right\}}_{n=1}^{N},\Pi \right)=q_{2}\left({\left\{w_{n}\right\}}_{n=1}^{N}\right)q_{2}\left(\{{\hat{t}}_{m},{\hat{w}}_{m}\}\right) \end{array}$$

Given Equation (Equation 17), we define the first factor as
$$\begin{array}{c} q_{2}^{*}\left(w_{n}\right)\propto \exp\left(-\frac{{\langle \phi_{n}^{2}\rangle }_{q_{1}^{*}}}{2}w_{n}\right)PG\left(w_{n}|1,0\right), \end{array}$$
which corresponds to a tilted PG distribution
(24)$$\begin{array}{c} q_{2}^{*}\left(w_{n}\right)=PG\left(w_{n}|1,\sqrt{{\langle \phi_{n}^{2}\rangle }_{q_{1}^{*}}}\right). \end{array}$$
with known expectations [21].

The second factor takes the form
$$\begin{array}{cc} \; & q_{2}^{*}\left({\left\{{\hat{t}}_{m},{\hat{w}}_{m}\right\}}_{m=1}^{M}\right)\\ \; & \propto \prod_{m=1}^{M}\exp\left(-\frac{{\langle \phi_{m}\rangle }_{q_{1}^{*}}}{2}-\frac{{\langle \phi_{m}^{2}\rangle }_{q_{1}^{*}}}{2}w_{m}\right)\cdot \exp\left({\langle \ln\lambda^{*}\rangle }_{q_{1}^{*}}\right). \end{array}$$

It can be shown that this distribution corresponds to a Poisson process with intensity function
(25)$$\begin{array}{cc} \; & \Lambda_{q_{2}}\left(\hat{t},\hat{w}\right)\\ \; & =\exp\left({\langle \ln\lambda \rangle }_{q_{1}^{*}}\right)\frac{\exp\left(-\frac{{\langle \phi \rangle }_{q_{1}^{*}}}{2}\right)}{2\cosh\left({\langle \phi^{2}\rangle }_{q_{1}^{*}}\right)}PG\left(w_{m}|1,\sqrt{{\langle \phi^{2}\rangle }_{q_{1}^{*}}}\right) \end{array}$$
where to simplify the notation we write ϕ instead of $\phi \left(\hat{t}\right)$.

The VI algorithm is summarized in Algorithm 1.
**Algorithm** **1:** NH–GPS Variational Inference.
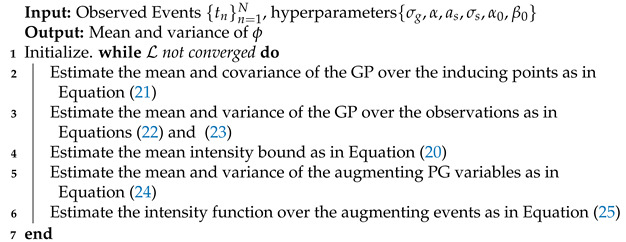


#### Hyperparameters Tuning

The Variational Inference does not tune the hyperparameters of the model. In order to achieve a better model, we wish to tune the hyperparameters of the model and to improve the likelihood of the model. Thus, we perform one step of gradient descent with respect to the ELBO at each iteration. The derivatives of the ELBO are given in Donner and Opper [19] (Appendix F). Notice that hyperparameter training, or tuning, is an implicit form of model selection, in that we are selecting among different models through the evaluation of the likelihoods.

### 3.5. Identifying the Background Rate and the Self-Effects Function

For some applications we may be interested in the specific shape of the background rate function *s* and the self-effects function *g*. At a first glance it is not entirely clear how to recover them from the inference, and we next describe how to do so.

Upon the convergence of the variational inference algorithm, we have an expression for the posterior mean and variance of the linear intensity function ϕ, as described in Equations (22) and (23). It is useful to define
(26)$$\tilde{g}\left(t\right)\;=\;g\left(t\right)\exp\left(-\alpha t\right)$$
and we similarly define
(27)$${\tilde{K}}^{g}\left(t,t^{\prime }\right)\;=\;K^{g}\left(t,t^{\prime }\right)\exp\left(-\alpha \left(t-t^{\prime }\right)\right).$$

We can rewrite the posterior mean in Equation (Equation 22) as
(28)$$\begin{array}{c} \langle \phi \left(t\right)\rangle \;=\;k_{c}^{s}{\left(t\right)}^{⊤}{\tilde{\mu }}_{c}+\sum_{l}\sum_{t_{i}<t}\sum_{t_{j}<t_{l}}{\tilde{K}}^{g}\left(t-t_{i},t_{l}-t_{j}\right){\tilde{\mu }}_{c}^{l} \end{array}$$
where we defined $\tilde{\mu_{c}}=K_{c}^{-1}\mu_{c}$ and used the fact that $k_{c}\left(t\right)$ can be written explicitly as
$$k_{c}^{s}\left(t\right)\;+\;\sum_{t_{i}<t}\sum_{t_{j}<t_{l}}{\tilde{K}}^{g}\left(t-t_{i},t_{l}-t_{j}\right)$$
with $k_{c}^{s}\left(t\right)$ being the kernel $K^{s}$ between data point *t* and all the inducing points.

From Equation (Equation 28) we can identify the posterior mean of *s* and *g* as
(29)$$\langle s\left(t\right)\rangle =k_{c}^{s}{\left(t\right)}^{⊤}{\tilde{\mu }}_{c}$$
(30)$$\langle g\left(t\right)\rangle =\sum_{l}\sum_{t_{j}<t_{l}}{\tilde{K}}^{g}\left(t,t_{l}-t_{j}\right){\tilde{\mu }}_{c}^{l}.$$

To identify the covariance of *s* and *g* we start with the posterior covariance of ϕ
(31)$$\begin{array}{cc} \mathrm{cov}\left(\phi \left(t\right),\phi \left(t^{\prime }\right)\right) & =\Sigma \left(t,t^{\prime }\right)=\tilde{K}\left(t,t^{\prime }\right)-k_{c}{\left(t\right)}^{⊤}Bk_{c}\left(t^{\prime }\right)\\ B & =K_{c}^{-1}-K_{c}^{-1}\Sigma_{c}K_{c}^{-1}. \end{array}$$

Using again the explicit form of $k_{c}\left(t\right)$ we rewrite the equation above as
(32)$$\begin{array}{cc} \Sigma \left(t,t^{\prime }\right)= & K_{t}^{s}+{\hat{K}}_{t}^{g}-{K_{tc}^{s}}^{⊤}BK_{tc}^{s}-2{K_{tc}^{s}}^{⊤}B{{\hat{K}}^{g}}_{tc}-{\hat{K}}_{tc}^{g⊤}B{{\hat{K}}^{g}}_{tc} \end{array}$$
where $K_{t}^{s/g}$ is the kernel matrix between the data points, $K_{tc}^{s/g}$ is the kernel between the data points and the set of inducing points, and we defined
$$\begin{array}{c} {\hat{K}}_{g}\left(t,t^{\prime }\right)=\sum_{t_{i}<t}\sum_{t_{j}<t^{\prime }}{\tilde{K}}_{g}\left(t-t_{i},t^{\prime }-t_{j}\right). \end{array}$$

From the expression above we can identify the marginal covariances of *s* and *g* separately, as well as their joint covariance. To sample *s* and $\tilde{g}$ from their posterior distribution we would need the full expression of the covariance, including both the marginal and cross covariances. For analyzing the model’s ability to recover *s* and $\tilde{g}$ we are interested in the marginal covariances, which can be expressed as
(33)$$\begin{array}{cc} \mathrm{cov}\left(s\left(t\right),s\left(t^{\prime }\right)\right) & =K_{t}^{s}-{K_{tc}^{s}}^{⊤}BK_{t^{\prime }c}^{s}\\ \mathrm{cov}\left(\tilde{g}\left(t\right),\tilde{g}\left(t^{\prime }\right)\right) & ={\tilde{K}}_{t}^{g}-\sum_{l,m}\sum_{\begin{array}{c} t_{j}<t_{l}\\ t_{i}<t_{m} \end{array}}{\tilde{K}}_{g}\left(t,t_{l}-t_{j}\right)B_{l,m}{\tilde{K}}_{g}\left(t^{\prime },t_{m}-t_{i}\right). \end{array}$$

## 4. Related Work

Before presenting the results of the experiment for our model, we go through recent developments in the field. We start with a short overview of Bayesian inference for Cox and Hawkes processes and then focus on the three models to which we later compare our approach.

Bayesian approaches to Cox processes model the intensity with a Gaussian process prior, which is then passed through a link function to ensure its positivity. A common choice of the link function is the exponential or the quadratic functions [27,28]. Another choice, which is more relevant to our work, is the sigmoid link function, resulting in the *sigmoidal Gaussian Cox process*. Inference in this model was first done empirically in study [29], as well as with moment-based parameters estimators [30]. Markov chain Monte Carlo methods were also developed [31], as well as variational inference [19].

In our work, we use a Bayesian semi-parametric inference approach. Earlier work, which introduces Bayesian nonparametric approaches to point processes, includes, among others, Ishwaran and James [32], who define kernel mixtures of Gamma measures for the intensity, Wolpert and Ickstadt [33], who define inhomogeneous Gamma random fields, and Taddy and Kottas [34], where joint nonparametric mixtures are introduced.

As for the Hawkes process, first attempts to perform Bayesian inference relied on the definition in terms of a marked Poisson cluster process and identifying the branching structure of the self-excitation [16]. One model that uses this approach is the mutually regressive point process (MRPP) [20]. MRPP is designed to model neuronal spike trains. In this work, the classical self-excitatory Hawkes Process intensity function is augmented by a probability term. This term induces inhibition when it is close to zero. In a sense, this model includes two memory kernels—one excitatory only, which appears in the intensity function, and another which can also induce inhibition in the augmenting probability term. Different from the MRPP, in our work, we achieve such flexibility of the self-effects in a simpler fashion by assuming the GP prior on the self-effects. As mentioned before, this also allows for the type of effect to change over time, which does not appear in the work of Apostolopoulou et al. [20].

A highly flexible approach to estimating the intensity function of the Hawkes process relies on GP priors [35,36,37]. A recent adaptation of this approach is the model described in Zhou et al. [17]. Similar to the model described in our work, the authors avoid the limiting parameterization of the memory kernel by using GPs. Different from our work, Zhou et al. [17] remain in the linear Hawkes process regime and assume that the effects of past events are only excitatory, whereas our approach allows both excitatory and inhibitory effects.

The last variation we describe is a sigmoid nonlinear multivariate Hawkes process (SNMHP) [38]. In this work, Zhou et al. describe a multivariate nonlinear HP, where, similarly to our work, the chosen link function is a sigmoid; however, we chose a nonparametric approach to model the causal influence function with a weighted mixture of basis functions from a certain family. Similar to the MRPP model, SNMHP was designed to model neuronal activity. A common assumption in this field is that each neuron is affected only by a subset of the other neurons in the network. Zhou et al. incorporate this assumption directly to their model by including a sparsity-inducing prior over the weights. In terms of inference, Zhou et al. proposed an expectation maximization algorithm, whereas we propose a fully Bayesian approach.

## 5. Experiments

In this section, we demonstrate the performance of our model and inference algorithm in different fields. First, we establish that the variational inference algorithm described in Section 3 is reasonable, using synthetic data as ground truth. Next, we apply the model and the inference algorithm to real-world datasets in the field of neuroscience and crime prediction. In these examples, the data are time series of events. We feed these events to the model and perform inference to estimate the underlying intensity function. The inferred intensity function can be used in different ways. For one, we use it to estimate the performance of the model (by calculating the log-likelihood or other metrics) and compare it to competing models. We also use it to simulate data from the model, which helps us assess whether the model is a reasonable candidate to describe the data (see Section 5.2.2). In the case of multivariate data, we use the inferred intensity function to assess the interaction between the different components of the data (see Section 5.2.3).

All the algorithms and experiments for this work are implemented in Python and are available online. To parallelize the computation over the available computing resources, we used the JAX package [39]. In the Gibbs sampler, the sampling of the PG variables was done using the PyPólyaGamma package [40]. The code and data are publicly available. See the Data Availability Statement.

### 5.1. Synthetic Data

To assess the performance of the inference algorithms presented in Section 3, we learn the parameters of the data generated by the model, namely the intensity function and the intensity bound, and compare the learned parameters to the ground truth. To generate data, we start by sampling the memory GP and the background GP, based on Equations (Equation 5) and (6). We generate events from the model using Poisson thinning [41]. First, we sample the number of candidates $J∼Poisson\left(\lambda T\right)$, and sample candidate events $\left\{t_{1},\ldots t_{J}\right\}$ uniformly. Next, we chronologically iterate through the candidates and accept them with probability $\frac{\Lambda \left(t_{j}|\left\{t_{1},\ldots t_{j-1}\right\}\right)}{\lambda }$.

The results for the synthetic data are included in Figure 1. The time window used was one second, and the dataset includes 91 events. In panel (a), the comparison between the ground truth predictive intensity and the one inferred by the learning algorithms demonstrates the accuracy of the inference methods. We compare the ground truth to the mean of the Gibbs samples, and the mean of the approximating distribution of the VI.

Panel (b) compares the ground truth value of the linear intensity and the one inferred by the two learning algorithms. In this example, the linear intensity is negative in some of the time points. Unlike for the predictive density, the Gibbs sampler is more accurate than the VI. Panel (c) presents the inference results for the intensity bound. As expected, the approximated distribution by the VI is much more narrow than the distribution of the Gibbs samples.

Panels (d) and (e) show the autocorrelation of the intensity bound and the ELBO through the Gibbs samples and VI iterations. In this example the convergence of the ELBO is very fast, and the autocorrelation of the Gibbs sample vanishes only after a few thousand iterations. We use the test log-likelihood per data point, averaged over ten datasets, to quantify the performance of the two inference algorithms. The Gibbs sampler and the VI achieve very similar results.

In the experiment presented above, the data were generated from the model, where *s* and *g* were sampled from a GP. When dealing with real data, we cannot expect that the data follow the model exactly. To demonstrate that our model can fit data that were not sampled from it directly, we infer the intensity when the data is generated from a slightly different model. In this case we take $s\left(t\right)=\beta_{1}\cos\left(\theta_{1}t\right)$ and $g\left(t\right)=\beta_{2}\cos\left(\theta_{2}t\right)$. The results can be found in Figure 2. We perform the same analysis presented in Figure 1. Similarly, both the Gibbs sampler and the VI algorithm recover the underlying intensity very well. Both inference methods achieve comparable results in terms of log-likelihood averaged over 10 test datasets. As the VI algorithm achieves similar results to the Gibbs sampler in a much faster computation time, in the next sections we present the results only for the VI algorithm.

In Section 3.5, we discussed the identifiability of the background rate function and the self-effects function from the inferred linear intensity. In Figure 3, we present the results of the inference of these functions from synthetic data. Both functions are recovered from the inference results of the linear intensity. In panel c, we can see that $\tilde{g}$ goes to zero for longer time differences, as expected from the integration of the exponential decay into the self-effects kernel.

It is important to notice that the stationarity conditions for nonlinear Hawkes processes are not sustained in our simulation procedure, as we ignore the past of the process, and time t<0. There are no events prior to the beginning of the simulation that will have self-effects. In this case, we generate a transient version rather than the stationary version of the process.

When *t* is big enough, we would expect to attain a stationary version of the process. However, the full conditions of stationarity for our process are not well-defined, and they are out of the scope of the current work. Nonetheless, conditioned on a given sample from the GP, to attain stationarity one only needs to ensure that $\int_{0}^{\infty }g\left(t\right)dt<\infty$ [8], as the intensity is bounded by λ.

Notably, our inference procedure is independent of the stationary nature of the driving GP, which might itself not be stationary, depending on the kernels used to define it. This implies that for these conditions to hold, we must study the process in a per-kernel basis.

### 5.2. Real Data

#### 5.2.1. Crime Report Data

Our model assumes both inhibitory and excitatory self-effects, but it should also be able to capture phenomena where only one of the two types of effects exist. To test this, we fit our model to crime report data, where it is assumed that past events have an excitatory effect on future events [42].

In criminology, it is known that crimes are clustered events. For example, burglars will repeatedly attack nearby targets in order to take advantage of known vulnerabilities, and gang-conflict shootings may incite or encourage retaliatory violence from the opposing gang in the local territory of the rivals. The nature of such retaliatory events, as well as the exploitation of resources by the burglars, are highly random and context-based phenomena, since this will depend on the criminals, location, or gang at hand. Hence, a highly flexible approach is required, and our methodology is able to express the inhomogeneity of each crime pattern through the unknown function *g* and *s*.

We use the same two datasets described in Zhou et al. [17], and follow their data processing procedure. Each dataset contains one type of crime, and so we use the univariate version of the model.

An example of the results of the inference process is presented in Figure 4. It includes the inferred linear intensity of the fitted model over the first 80 events in the Vancouver crime dataset.

Table 1 compares the test log-likelihood of our NH–GPS model to the one reported in Zhou et al. [17]. The work of Zhou et al. [17] includes several inference methods and we compare our results to the results of their reported mean-field variational inference approach as it is the closest to our inference procedure. We perform the experiment five times and report the mean and variance of the test log-likelihood. As expected, our model performs similarly to the semi-parametric Hawkes process presented by Zhou et al. [17].

#### 5.2.2. Neuronal Activity Data

One of the motivating real-world phenomena behind our work is the spiking activity of neurons, where it is known that the process has both self-excitatory and self-inhibitory effects. Having a reliable model that captures the data accurately is useful to identify and analyze different physiological mechanisms that drive the neuronal activity.

As an example for our model’s ability to capture neuronal activity we use the datasets that were first presented in Gerhard et al. [43] (Figure 2b,c). One dataset includes ten recordings from a single neuron in a monkey cortex, with a duration of one second each, and the other includes ten recordings from a single neuron in a human cortex for a duration of ten seconds each. In this work, point process generalized linear models were used to fit the data, and the data generated from it yielded unrealistic spiking patterns. This hints the need for a nonlinear model to describe the data.

The dataset described above were further analyzed in Apostolopoulou et al. [20] (Figure 5), where the mutually regressive point process (MR–PP) is introduced. The fitted MR–PP model produced realistic spike trains and we use it as a comparison for our model.

We fit the model to the multitrial single neuron datasets and infer the intensity of the assumed underlying point process. Figure 5 presents the results for one trial from each dataset. In both cases, the inferred linear intensity obtains both positive and negative values, which implies both excitatory and inhibitory spiking patterns.

In Figure 6, we assess the ability of the model to capture the data. The left column includes the raster plot of the real data and the middle plot the raster plot generated from the fitted model. Similarly to the real data, the generated data displays both excitation, in the form of clustered events, and inhibition.

To quantify how suitable the model is to the data, we apply the random time change theorem [18] to the inferred intensity and the experimental data. The theorem states that realizations from a general point process can be transformed to realizations from a homogeneous Poisson process with a unit rate. Similarly to the work of Apostolopoulou et al. [20], we further transform the exponential realizations to those from a uniform distribution, following Brown et al. [44]. We then use the Kolmogorov–Smirnov test to compare the quantiles of the distribution of the transformed realizations to the quantiles of the uniform distribution. The results of this test are displayed in Figure 6 in the right column. The comparison relies on 95% confidence bounds, which are indicated by the dashed lines. The model passes the goodness-of-fit test (*p*-value >0.05), and the *p*-value is higher than the one achieved by the MR–PP. We compare the reported *p*-value achieved by the MR–PP model for the two datasets in Table 2. For both datasets, our model achieves higher a *p*-value than the MR–PP.

#### 5.2.3. Multi-Neurons Data

Last, we demonstrate the performance of the multivariate version of our model. We use the data presented in Zhou et al. [38]. This dataset includes spike trains simultaneously recorded from 25 neurons in the primary visual cortex of an anesthetized cat.

One application of our model for the use-case of multi-neuron recording, is the analysis of the interactions between the different neurons. As presented in Section 3.5, after fitting the model to the data we can recover the function *g*, which describes the effects of past events on future events. In the case of a multivariate model, this function is defined for every pair of dimensions, and describes the interaction between them. An example of this can be found in Figure 7. On the left is the recorded data, and on the right is the interactions between two neurons from the dataset.

Next, we compare the performance of our model to the model described in Zhou et al. [38], by looking at the test log-likelihood.. We follow the same train-test split as Zhou et al. [38], where the first 30 seconds of the recording are used as the training set and the last 30 seconds are used as the test set.

The test log-likelihood can be found in Table 3. We compare our model the the one from Apostolopoulou et al. [20] (MR–PP) and the model presented in Zhou et al. [38] (SNMHP). Our model achieves a higher test log-likelihood score than the competing models. This demonstrates the power in the flexibility of our model and showcases its applicability also to multivariate data.

## 6. Discussion

In this work, we present the nonlinear Hawkes model with Gaussian process self-effects (NH–GPS) and a Bayesian variational inference scheme to infer its parameters. We motivated the development of the new model with the need for a flexible model that can capture both exciting and inhibiting interactions between events, while maintaining the ability to learn also when data are scarce. The model includes both a univariate and a multivariate version.

Due to the structure of the model, we derive an inference algorithm without the branching structure that is commonly used for Bayesian inference in Hawkes processes. We propose a mean-field variational inference algorithm, which relies on a data augmenting scheme. We show that the results of the variational inference are comparable with those of a Gibbs sampler.

We demonstrate the performance of our model in four different real-world applications. Due to the flexibility of our model, it achieves good results on data where events have only excitatory effects and on data where events have both excitatory and inhibitory effects.

Different from recent work on nonlinear Hawkes process inference [38], our work presents a two-fold advantage. The introduction of the Gaussian Process allows for a wider range of applications as well as the Bayesian perspective, which lends itself to uncertainty quantification, model selection, and regularization. All thought that nonlinear Hawkes processes were ubiquitous in applications for finance, social dynamics, and neuroscience. Previous works are tailored with specific applications in mind, whereas our methodology is of a general nature.

More importantly, the introduction of priors, as well as the GP, allows the practitioner to easily introduce expert knowledge, as one is able to modify the behavior of the self-effects by introducing different kernels.

In this work, we did not include results regarding the prediction abilities of the model, as we leave it for future work. We do believe it is relatively simple to acquire in our model and we briefly describe how to do so. The inference over the aggregated history function ϕ, which includes both the sum of the Gaussian process associated with the exogenous arrival rate and the endogenous event rate, allows us to generate estimates for predictions. Once we sample the value of ϕ conditioned on the previous observed events, we can estimate the mean arrival time of the next event by estimating the integral $E\left(t_{n+1}\right)=\int_{0}^{\infty }tP\left(t|H\right)$ via numerical methods, such as Monte Carlo integration.

We would also like to expand our model to describe its spatio-temporal processes. Our model can be directly extended to capture events in time and space, as the memory kernel is reduced to a kernel function of a GP, which can be applied also to multi-dimensional data.

## Figures and Tables

**Figure 1 entropy-24-00356-f001:**
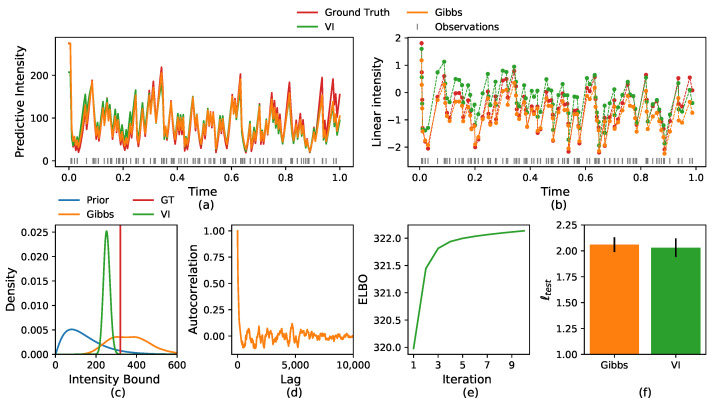
(**a**) Comparison of the ground truth predictive intensity and the one sampled from the VI and Gibbs inference. (**b**) Comparison of the ground truth linear intensity $\phi \left(\cdot \right)$ and the one learned by the VI and Gibbs sampler. (**c**) Comparison of the ground truth intensity bound and the one learned by the inference, and the prior distribution. (**d**) The autocorrelation of the intensity bound Gibbs samples. (**e**) The variational lower bound as a function of the algorithm iteration. (**f**) Comparison of the test log-likelihood of the Gibbs sampler and the VI.

**Figure 2 entropy-24-00356-f002:**
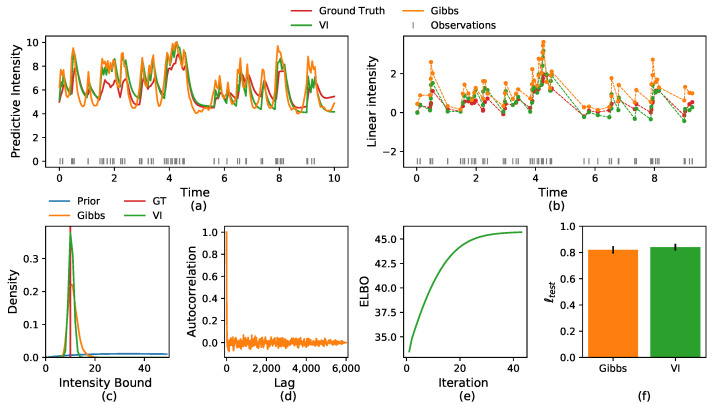
(**a**) Comparison of the ground truth predictive intensity and the one sampled from the VI and Gibbs inference. (**b**) Comparison of the ground truth linear intensity $\phi \left(\cdot \right)$ and the one learned by the VI and Gibbs sampler. (**c**) Comparison of the ground truth intensity bound and the one learned by the inference, and the prior distribution. (**d**) The autocorrelation of the intensity bound Gibbs samples. (**e**) The variational lower bound as a function of the algorithm iteration. (**f**) Comparison of the test log-likelihood of the Gibbs sampler and the VI.

**Figure 3 entropy-24-00356-f003:**
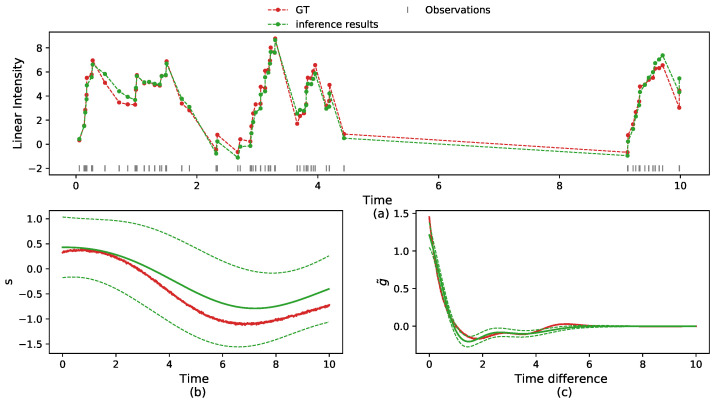
(**a**) Comparison of the ground truth linear intensity and the one inferred by the VI algorithm. (**b**) Comparison of the ground truth background rate function *s* and the one inferred by the VI algorithm. (**c**) Comparison of the ground truth self-effects function *g* and the one inferred by the VI algorithm.

**Figure 4 entropy-24-00356-f004:**
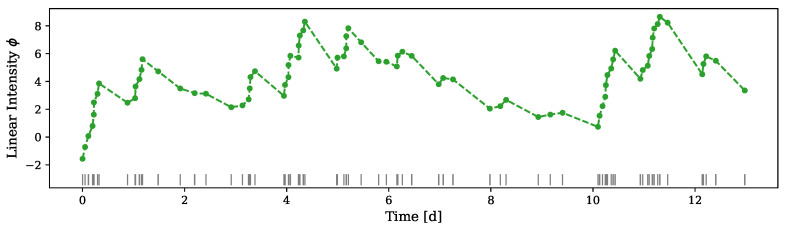
Inferred linear intensity ϕ for the first 80 events in the Vancouver crime dataset. The linear intensity is estimated only for the observed events and the dashed line between the dots is merely linear interpolation.

**Figure 5 entropy-24-00356-f005:**
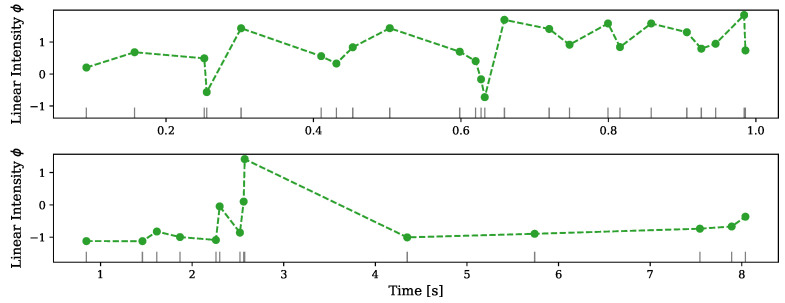
Inferred intensity for single neuron data. Upper panel—results for trial number 9 from the dataset recorded from monkey cortex. Lower panel—results for trial number 7 from the dataset recorded from human cortex. The linear intensity is estimated only for the observed events and the dashed line between the dots is merely linear interpolation.

**Figure 6 entropy-24-00356-f006:**
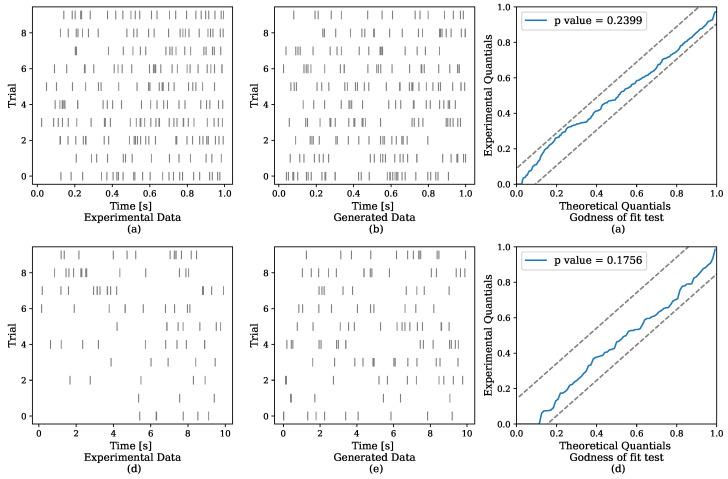
**Upper row**—single neuron recordings from monkey cortex. **Lower row**—single neuron recordings from human cortex. **Left column**—recorded data. **Middle column**—data generated from the learned models. **Right column**—results of the Kolmogorov–Smirnov test. The NH–GPS generates data that resembles the real data, and passes the goodness of fit test.

**Figure 7 entropy-24-00356-f007:**
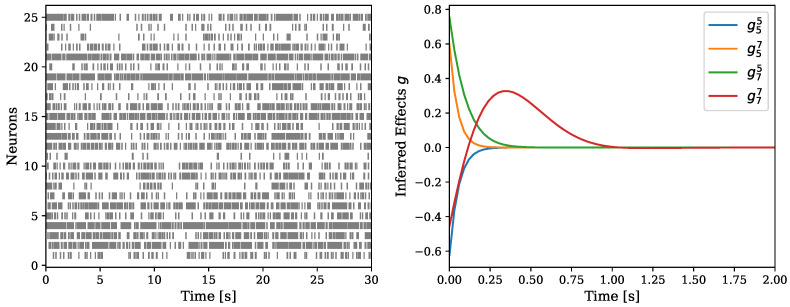
(**Left**)—the recorded activity of 25 neurons. (**Right**)—example of the recovery of the influence function *g* between neuron number 5 and neuron number 7. Both excitatory and inhibitory influence is observed.

**Table 1 entropy-24-00356-t001:** Crime report data test log-likelihood.

Dataset	Zhou et al. (2020) [17]	NH–GPS
Vancouver	453.11±8.94	453.8±12.2
NYPD	−200.7±3.32	−202.8±7.54

**Table 2 entropy-24-00356-t002:** *p*-Value of the KS Test on neuronal activity data.

Dataset	MR–PP	NH–GPS
Monkey Cortex	0.103	0.23
Human Cortex	0.096	0.175

**Table 3 entropy-24-00356-t003:** Test log-likelihood for the multi-neurons datasets for different models.

SNMHP	MR–PP	NH–GPS
$-6.13\times 10^{3}$	$-5.9\times 10^{3}$	$-5.29\times 10^{3}$

## Data Availability

All the code and data included in this paper can be found at https://github.com/noashin/NHGPS (accessed on 28 January 2022).

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
