# Peer review of "Variational Bayesian Inference for Nonlinear Hawkes Process with Gaussian Process Self-Effects"

_entropy, 2022, doi:10.3390/e24030356_

Round 1

Reviewer 1 Report

See my report

Reviewer 2 Report

1. Upon first reading this paper, It is hard to imagine model(2-4) having any kind of physical meaning or basis, and I questioned whether it would fit well to actual data. However, I appreciate the authors' honesty in Table 1, showing that their model fits slightly better in Vancouver and slightly worse in NYPD. To me, such nuanced results are more informative than the typical paper that tries in every way to show only results supporting the optimality of the present model. I also appreciate the use of time rescaled residuals as a form of model evaluation. This is great. However, all the applications to data appear to be a bit cursory, and one does not finish reading the paper with a clear sense of what the data or the fitted intensities look like. I suggest the authors include a few more plots indicating what the data and the fitted intensities look like, in each case. 

  2. I also wonder if other types of residual plots and tests might be worth investigating, like superthinned residuals, and tests like the L-test or R-test, as for example in Clements et al. (2011), Zechar et al. (2013), or Gordon et al. (2015)?  Clements, R.A., Schoenberg, F.P., and Schorlemmer, D. (2011). Residual analysis for space-time point processes with applications to earthquake forecast models in California. Annals of Applied Statistics 5(4), 2549--2571. Gordon, J.S., Clements, R.A., Schoenberg, F.P., and Schorlemmer, D. (2015). Voronoi residuals and other residual analyses applied to CSEP earthquake forecasts. Spatial Statistics , 14b, 133-150. ZECHAR, J. D., SCHORLEMMER, D., WERNER, M. J., GERSTENBERGER, M. C., RHOADES, D. A. and JORDAN, T. H. (2013). Regional earthquake likelihood models I: First-order results. Bulletin of the Seismological Society of America, 103, 2A, 787-798.    3. As unfortunately usual in Bayesian papers, there is no justification for the priors, no assessment of sensitivity to those priors, and no investigation into alternative priors. I would like to see some analysis of this issue.    4. In 5.2.2., in the figures, be sure to say "Goodness" not "Godness"! This means something quite different!   

Round 2

Reviewer 1 Report

See my report

Reviewer 2 Report

The authors did a satisfactory job addressing my concerns. I still think the article can be substantially improved with some model evaluation, but as it is I consider it good enough for publication.